# When do random forests fail?

**Cheng Tang**
George Washington University
Washington, DC
tangch@gwu.edu

**Damien Garreau**
Max Planck Institute for Intelligent Systems
Tübingen, Germany
damien.garreau@tuebingen.mpg.de

**Ulrike von Luxburg**
University of Tübingen
Max Planck Institute for Intelligent Systems
Tübingen, Germany
luxburg@informatik.uni-tuebingen.de

## Abstract

Random forests are learning algorithms that build large collections of random trees and make predictions by averaging the individual tree predictions. In this paper, we consider various tree constructions and examine how the choice of parameters affects the generalization error of the resulting random forests as the sample size goes to infinity. We show that subsampling of data points during the tree construction phase is important: Forests can become inconsistent with either no subsampling or too severe subsampling. As a consequence, even highly randomized trees can lead to inconsistent forests if no subsampling is used, which implies that some of the commonly used setups for random forests can be inconsistent. As a second consequence we can show that trees that have good performance in nearest-neighbor search can be a poor choice for random forests.

## 1   Introduction

Random forests (Breiman, 2001) are considered as one of the most successful general-purpose algorithms in modern-times (Biau and Scornet, 2016). They can be applied to a wide range of learning tasks, but most prominently to classification and regression. A random forest is an ensemble of trees, where the construction of each tree is random. After building an ensemble of trees, the random forest makes predictions by averaging the predictions of individual trees. Random forests often make accurate and robust predictions, even for very high-dimensional problems (Biau, 2012), in a variety of applications (Criminisi and Shotton, 2013; Belgiu and Drăguţ, 2016; Díaz-Uriarte and Alvarez de Andrés, 2006). Recent theoretical works have established a series of consistency results of different variants of random forests, when the forests' parameters are tuned in certain ways (Scornet, 2016; Scornet et al., 2015; Biau, 2012; Biau et al., 2008). In this paper, however, we ask the question of when do random forests *fail*. In particular, we examine how varying several key parameters of the algorithm affects the generalization error of forests.

When building a random forest, there are several parameters to tune: the choice of the base trees (the randomized algorithm that generates the individual trees), the number of trees in the forest, the size of the leaf nodes, the rate of data subsampling, and sometimes the rate of feature subsampling. Popular variants of random forests usually come with their own default parameter tuning guidelines, often suggested by practice. For example, common wisdom suggests that training a large number of trees and growing *deep* trees whose leaf sizes are fixed to a small constant lead to better performance. For data subsampling, the original random forest paper (Breiman, 2001) suggests to set the subsampling (with replacement) rate to be 1, while a later popular variant (Geurts et al., 2006)

proposes to disable data subsampling altogether. For feature subsampling, the consensus is to set the rate to $d/3$ for regression problems, with $d$ being the dimension (Friedman et al., 2009, Section 15.3). But in Díaz-Uriarte and Alvarez de Andrés (2006), the feature sampling rate is found to be not important, while Genuer et al. (2010) suggests to not subsample the features.

Existing analyses of random forests mostly focus on positive results and typically fall into two categories: (1) They show a forest is consistent by showing that its base trees are consistent (Biau et al., 2008; Biau, 2012; Denil et al., 2014). This class of results does not cover the case of deep trees (because individual deep trees are clearly inconsistent), and fails to highlight the advantage of using random forests as opposed to single trees. (2) In the deep tree regime, recent theoretical consistency results require subsampling as a *sufficient* condition for consistency (Scornet, 2016).

We focus on negative results: When are random forests inconsistent? To facilitate our theoretical investigation, we restrict our analysis to unsupervised random forests, that is, random forests whose tree construction does not use label information (Def. 2). We establish two conditions, *diversity* and *locality* (Def. 3 and 4), that are *necessary* for a forest to be consistent. We then examine how parameter tuning affects diversity and locality. Our results highlight the importance of subsampling data points during the tree construction phase: Without subsampling, forests of deep trees can become inconsistent due to violation of diversity; on the other hand, if we subsample too heavily, forests can also become inconsistent due to violation of locality. Our analysis implies two surprising consequences as special cases: (1) When considering partitioning trees that are particularly good for nearest-neighbor search, such as random projection trees, it is natural to expect them to be also good for random forests. Our results disagree with this intuition: Unless we use severe subsampling, they lead to inconsistent forests. (2) In a popular variant of random forests, extremely randomized trees are used and subsampling is disabled (Geurts et al., 2006). The argument in that paper is that when forests use extremely randomized trees, the randomness in the trees already reduces variance and thus subsampling becomes unnecessary. Our results suggest otherwise.

## 2  Background on random forests

Throughout this paper, we consider $n$ i.i.d. samples $X_1, \ldots, X_n$ of an unknown random variable $X$ that has support included in $[0,1]^d$. Let $\eta : [0,1]^d \to \mathbb{R}$ be a measurable function. The responses $Y_1, \ldots, Y_n$ are $\mathbb{R}$-valued random variables which satisfy

$$\forall 1 \leq i \leq n, \quad Y_i = \eta(X_i) + \varepsilon_i, \tag{2.1}$$

where the $\varepsilon_i$ are centered random variables with variance $\sigma^2 > 0$. We assume that they are independent from the observations. For any integer $n$, we set $[n] := \{1, \ldots, n\}$. We denote by $X_{[n]} := (X_i)_{1 \leq i \leq n}$ the training set, $Y_{[n]} := (Y_i)_{1 \leq i \leq n}$ the responses, and $\mathcal{D}_n := (X_i, Y_i)_{1 \leq i \leq n}$ the training sample. We focus on the regression problem, that is, the problem of estimating the unknown regression function $\eta(x) = \mathbb{E}[Y|X = x]$ by constructing an estimator $\widehat{\eta}_n(x)$ based on the training sample $\mathcal{D}_n$. We define the mean squared error of any estimator $\widehat{\eta}_n$ as $\mathbb{E}\left[|\widehat{\eta}_n(X) - \eta(X)|^2\right]$, and we say that the estimator is $\mathbb{L}^2$-*consistent* if the mean squared error goes to zero when the sample size grows to infinity, that is,

$$\lim_{n \to \infty} \mathbb{E}\left[|\widehat{\eta}_n(X) - \eta(X)|^2\right] = 0. \tag{2.2}$$

The present paper examines the consistency of random forests as estimators of the regression function. Here and in the rest of this article the expectation $\mathbb{E}[\cdot]$ is taken with respect to the random variables $X, X_1, \ldots, X_n, \varepsilon_1, \ldots, \varepsilon_n$, and any additional source of randomness coming from the (random) tree construction, unless otherwise specified.

**Regression trees.** A random forest makes predictions by aggregating the predictions of tree-based estimators. To obtain a tree-based estimator, one first uses the training sample to build a "spatial partitioning tree." Any query $x$ in the ambient space is then routed from the root to a unique leaf node and assigned the mean value of the responses in the corresponding cell.

Formally, the $j$-th tree in the ensemble constructed from training sample $\mathcal{D}_n$ induces a hierarchy of finite coverings of the ambient space $[0,1]^d$: let $k$ denote the height of the tree. Then at every level $\ell \in [k]$ the tree induces a $p_\ell$-covering of the ambient space, namely subspaces $\mathbb{A}_1^j, \ldots, \mathbb{A}_{p_\ell}^j \subset$

$[0,1]^d$ such that $\bigcup_{i=1}^{p_\ell} \mathbb{A}_i^j = [0,1]^d$. Each cell $\mathbb{A}_i^j$ corresponds to a node of the tree. The tree-induced routing of a query to a unique cell in space at level $\ell \in [k]$ is a function $A_\ell^j : [0,1]^d \to \{\mathbb{A}_1^j, \ldots, \mathbb{A}_{p_\ell}^j\}$; it satisfies $\forall x \in [0,1]^d, \exists! \ i \in \{1, \ldots, p_\ell\}$ such that $A_\ell^j(x) = \mathbb{A}_i^j$. In the following, we refer to function $A_\ell^j$ as the *routing function* associated with tree $j$ at level $\ell$, and we will often identify the trees with their associated functions at level $k$, $A_k^j$ (or simply $A^j$ when there is no ambiguity). Note that this routing function is well-defined even for tree structures that allow overlapping cells.

Once a tree $A^j$ has been constructed, it estimates the regression function $\eta(x)$ for a query point $x$, using only information on training points contained in cell $A^j(x)$. Formally, given a query point $x$ let $N(A^j(x))$ denote the number of samples that belong to the cell $A^j(x)$. We define the $j$-th tree-based estimator $\widehat{\eta}_{n,A^j} : [0,1]^d \to \mathbb{R}$ as

$$\widehat{\eta}_{n,A^j}(x) := \frac{1}{N(A^j(x))} \sum_{i=1}^n Y_i \, \mathbb{1}_{X_i \in A^j(x)} \,,$$

with the convention $\frac{0}{0} = 0$. Intuitively, $\widehat{\eta}_{n,A^j}(x)$ is the empirical average of the responses of sample points falling in the same cell as $x$ — see Fig. 1. We refer to Friedman et al. (2009, Section 9.2.2) for a more detailed overview of regression trees.

**Random forests.** A random forest builds an ensemble of $T$ tree estimators that are all constructed based on the same data set and the same tree algorithm, which we call the *base tree algorithm*. Due to the inherent randomness in the base tree algorithm, which we denote by $\Theta$, each tree $A^j$ will be different; $A^j$ can depend on both the training data $\mathcal{D}_n$, and $\Theta$. For instance, the random variable $\Theta$ may encode what feature and threshold are used when splitting a node. An important source of randomness is the one coming from what we simply call "subsampling": when building each tree $A^j$, we do not use the entire data set during tree construction, but just a susbsample of the data (which can be with or without replacement). This source of randomness is also encoded by $\Theta$. Formally, the random forest estimator associated to the collection of trees $\mathbb{V}_T = \{A^j, 1 \leq j \leq T\}$ is defined by

$$\widehat{\eta}_{n,\mathbb{V}_T}(x) := \frac{1}{T} \sum_{j=1}^T \widehat{\eta}_{n,A^j}(x) = \frac{1}{T} \sum_{j=1}^T \frac{1}{N(A^j(x))} \sum_{i=1}^n Y_i \, \mathbb{1}_{X_i \in A^j(x)} \,. \tag{2.3}$$

We refer to Friedman et al. (2009, Chapter 15) and Biau and Scornet (2016) for a more comprehensive introduction to random forests algorithms.

**Local average estimators and infinite random forests.** An important fact about random forest estimators is that they can be seen as *local average estimators* (Devroye et al., 1996, Section 6.5), a concept that generalizes many nonparametric estimators, including histogram, kernel, nearest-neighbor, and tree-based estimators. A local average estimator takes the following generic form:

$$\widehat{\eta}_n(x) = \sum_{i=1}^n W_{n,i}(x) Y_i \,. \tag{2.4}$$

For a given query $x$, a local average estimator predicts its conditional response by averaging the responses in the training sample that are "close" to $x$. $W_{n,i}(x)$ can be thought of as the "weight" or the contribution of the $i$-th training point in predicting the response value for $x$.

Random forests form a special class of local average estimators: introducing the weights $W_{n,i}^T(x) := \frac{1}{T} \sum_{j=1}^T \frac{\mathbb{1}_{X_i \in A^j(x)}}{N(A^j(x))}$, we can immediately see from Eq. (2.3) that

$$\widehat{\eta}_{n,\mathbb{V}_T}(x) = \sum_{i=1}^n \frac{1}{T} \sum_{j=1}^T \frac{1}{N(A^j(x))} \mathbb{1}_{X_i \in A^j(x)} Y_i = \sum_{i=1}^n W_{n,i}^T(x) Y_i \,. \tag{2.5}$$

It is clear that the weights defined by a random forest are non-negative. To analyze the asymptotic properties of forests, there are different regimes that one can consider: the regime "fixed $T$, and

large $n$" essentially does not differ from analyzing an individual tree. To see advantages of forests, one needs to let both $T$ and $n$ go to infinity. As it is common in the literature on random forests, we first let $T \to \infty$ to get rid of the randomness $\Theta$ that is inherent to the tree construction: According to the law of large numbers, the estimator defined by Eq. (2.5) behaves approximately as an *infinite random forest* with associated estimator

$$\widehat{\eta}_{n,\mathbb{V}_\infty}(x) := \sum_{i=1}^{n} W_{n,i}^\infty(x) Y_i \,,$$

where $W_{n,i}^\infty(x) := \mathbb{E}_\Theta \left[ \frac{\mathbb{1}_{X_i \in A(x)}}{N(A(x))} \right]$ are the asymptotic weights and $A(\cdot)$ is the routing function associated with a generic random tree. Indeed, Scornet (2016, Theorem 3.1) shows that $\widehat{\eta}_{n,\mathbb{V}_\infty}(\cdot)$ is the limiting function of $\widehat{\eta}_{n,\mathbb{V}_T}(\cdot)$ as the number of trees $T$ goes to infinity. The concept of the infinite forest captures the common wisdom that one should use many trees in random forests (see the next paragraph). In the following, we focus on such infinite random forests. Now our question becomes: If we construct infinitely many trees by a particular base tree algorithm, is the forest consistent as the number $n$ of data points goes to infinity?

**Common beliefs and parameter setups in random forests.** Different variants of random forests usually have different parameter tuning principles. However, there are three common beliefs about random forests in general, both in the literature and among practitioners. The first belief is that *"many trees are good,"* in the sense that adding trees to the ensemble tends to decrease the generalization error of random forests (Biau and Scornet, 2016, Sec. 2.4). For example, the results in Theorem 3.3 of Scornet (2016) and Arlot and Genuer (2014) both corroborate this belief. The second belief is that, in the context of random forests, *"it is good to use deep trees"* (Breiman, 2000).

**Definition 1** (**Deep trees and fully-grown trees**)**.** *We say a random forest has deep trees if there exists an integer $n_0$ such that, for any sample size $n$, the leaf nodes of its base trees have at most $n_0$ points almost surely; a fully-grown tree is a deep tree whose leaves have exactly one data point.*

The use of deep trees seems counter-intuitive at first glance: They have low bias but extremely high variance that does not vanish as the sample size increases, and thus are destined to *overfit*. However, while a single deep tree estimator is clearly not consistent in general, it is believed that combining many deep trees can effectively reduce the variance of individual trees. Thus, it is believed that a random forest estimator takes advantage of the low bias of individual deep trees while retaining low variance. Recent work of Scornet (2016) provided theoretical evidence of this belief by showing that forests of fully-grown quantile trees are consistent under certain sufficient conditions. The third belief is that a diverse portfolio of trees helps alleviate overfitting (by reducing variance), and that *randomizing the tree construction helps creating a more diverse portfolio.* Since the introduction of random forest, "tree diversity," which has been defined as correlation of fit residuals between base trees in Breiman (2001), has been perceived as crucial for achieving variance reduction. It has also become a folklore knowledge in the random forest community that by introducing "more randomness," trees in the ensemble become more diverse, and thus less likely to overfit. In practice, many ways of injecting randomness to the tree construction have been explored, for example random feature selection, random projection, random splits, and data subsampling (bootstrapping). Geurts et al. (2006) suggest using *extremely randomized trees*; taking this idea to the limit yields the *totally randomized trees*, that is, trees constructed without using information from the responses $Y_{[n]}$. Our analysis takes into account all three common beliefs, and studies forest consistency under two extreme scenarios of subsampling setup.

## 2.1 Related Work

Random forests were first proposed by Breiman (2001), where the base trees are chosen as Classification And Regression Trees (CART) (Breiman et al., 1984) and subsampling is enabled during tree construction. A popular variant of random forests is called "extremely randomized trees" (extra-trees) (Geurts et al., 2006). Forests of extra-trees adopt a different parameter setup than Breiman's forest: They disable subsampling and use highly randomized trees as compared to CART trees. Besides axis-aligned trees such as CART, oblique trees (trees with non-rectangular cells) such as random projection trees are also used in random forests (Ho, 1998; Menze et al., 2011; Rodriguez et al., 2006; Tomita et al., 2015). On the theoretical side, all previous works that we are aware of investigate forests with axis-aligned base-trees. Most works analyze trees with UW-property (see Def. 2)

and focus on establishing consistency results (Scornet (2016); Biau (2012); Biau et al. (2008)). A notable breakthrough was Scornet et al. (2015), who were the first to establish that Breiman's forest, which do not satisfy the UW-property (Def. 2), is consistent on additive regression models. To our knowledge, few works focus on negative results. An exception is Lin and Jeon (2006), which provides a lower bound on the mean squared error convergence rate of forests.

## 2.2 Overview of our results

Section 3 establishes two notions, "diversity" and "locality," that are necessary for local average estimators to be consistent. Then, viewing infinite random forests as local average estimators, we establish a series of inconsistency results in Section 4. In Section 4.1, we show that forests of deep trees with either nearest-neighbor-preserving property (Def. 6) or fast-diameter-decreasing property (see condition in Prop. 1) violate the diversity condition, when subsampling is disabled. As a surprising consequence, we show that trees with nearest-neighbor-preserving property (Algorithm 1 and 2) can be inconsistent if we follow a common forest parameter setup (Def. 5). In Section 4.2, we show that when undersampled, forests of deep trees can violate the locality condition. Our analysis applies to trees that are both axis-aligned and irregularly shaped (oblique).

## 3 Inconsistency of local average estimators

A classical result of Stone (1977, Theorem 1) provides a set of *sufficient* conditions for local average estimators to be consistent. In this section, we derive new inconsistency results for a general class of local average estimator satisfying an additional property, often used in theoretical analyses:

**Definition 2** (**UW-property**). *A local average estimator defined as in Eq.* (2.4) *satisfies the "unsupervised-weights" property (UW-property) if the weights $W_{n,i}$ depend only on the unlabeled data.*

### 3.1 Diversity is necessary to avoid overfitting

We first define a condition on local average estimators, which we call *diversity*, and show that if local average estimators do not satisfy diversity, then they are inconsistent on data generated from a large class of regression models. In fact, from the proof of Lemma 1, it can be seen that violating diversity results in high asymptotic variance, hence inconsistent estimators.

**Definition 3** (**Diversity condition**). *We say a local average estimator as defined in Eq* (2.4) *satisfies the diversity condition, if $\mathbb{E}\left[\sum_{i=1}^{n} W_{n,i}^{2}(X)\right] \longrightarrow 0$ as $n \to \infty$.*

Intuitively, the diversity condition says that no single data point in the training set should be given too much weight asymptotically. The following lemma shows that diversity is necessary for a local average estimator (with UW-property) to be consistent on a large class of regression models.

**Lemma 1** (**Local average estimators without diversity are inconsistent**). *Consider a local average estimator $\widehat{\eta}_n$ as in Eq.* (2.4) *that satisfies the UW-property. Suppose the data satisfies Eq.* (2.1)*, and $\sigma$ be as defined therein. Suppose the diversity condition (Def. 3) is not satisfied: that is, there exists $\delta > 0$ such that $\mathbb{E}\left[\sum_{i=1}^{n} W_{n,i}^{2}(X)\right] \geq \delta$ for infinitely many $n$. Then $\widehat{\eta}_n$ is not consistent.*

A related result is proved in Stone (1977). It considers the artificial scenario where the data distribution $(X, Y)$ satisfies that (i) $Y$ is *independent* of $X$, and (ii) $Y$ is standard Gaussian. On this particular distribution, Stone (1977, Prop. 8) shows that condition (5) of Stone (1977, Theorem 1) is necessary for a local average estimator to be consistent. In contrast, our Lemma 1 applies to a much larger class of distributions.

### 3.2 Locality is necessary to avoid underfitting

Now we introduce another necessary condition for the consistency of local average estimators, which we call *locality*. While diversity controls the variance of the risk, locality controls the bias.

**Definition 4** (**Locality condition**). *We say that a local average estimator $\widehat{\eta}_n$ with weights $W_{n,i}$ satisfies the locality condition if, for any $a > 0$, $\mathbb{E}\left[\sum_{i=1}^{n} W_{n,i}(X)\,\mathbb{1}_{\|X_i - X\| > a}\right] \longrightarrow 0$ as $n \to \infty$.*

The locality condition is one of the conditions of Stone's theorem for the consistency of local average estimators. In plain words, it requires the estimator to give small weight to sample points located

| **Algorithm 1** Randomized Projection Tree | **Algorithm 2** Randomized Spill Tree |
|---|---|
| **Input:** Sample $S$, maximum leaf size $n_0$; | **Input:** $S, n_0, \alpha \in (0, 1/2)$; |
| **Output:** $T = RPT(S, n_0)$; | **Output:** $T = RST(S, n_0, \alpha)$; |
| 1: $T \leftarrow$ empty tree; | 1: $T \leftarrow$ empty tree; |
| 2: **if** $\|S\| > n_0$ **then** | 2: **if** $\|S\| > n_0$ **then** |
| 3:     Sample $U$ uniformly from $S^{d-1}$; | 3:     Sample $U$ uniformly from $S^{d-1}$; |
| 4:     Sample $q$ uniformly from $\left[\frac{1}{4}, \frac{3}{4}\right]$; | 4:     $t_L \leftarrow$ top $\frac{1}{2} + \alpha$-quantile of $U^T \cdot S$; |
| 5:     $t_q \leftarrow$ empirical $q$-th quantile of $U^T \cdot S$; | 5:     $t_R \leftarrow$ bottom $\frac{1}{2} + \alpha$-quantile of $U^T \cdot S$; |
| 6:     $S_L \leftarrow \{x \in S : U^T \cdot x \le t_q\}$; | 6:     $S_L \leftarrow \{x \in S : U^T \cdot x \le t_L\}$; |
| 7:     $T.\text{graft}\,(RPT(S_L, n_0))$; | 7:     $T.\text{graft}\,(RST(S_L, n_0, \alpha))$; |
| 8:     $S_R \leftarrow S \setminus S_L$; | 8:     $S_R \leftarrow \{x \in S : U^T \cdot x \ge t_R\}$; |
| 9:     $T.\text{graft}\,(RPT(S_R, n_0))$; | 9:     $T.\text{graft}(RST(S_R, n_0, \alpha))$; |
| 10: **end if** | 10: **end if** |

outside a ball of fixed radius centered around a query. Indeed, intuitively, a local average estimator should be able to capture fine-scale changes in the distribution of $X$ in order to be consistent. Our next result shows that there exists a distribution such that, when a local average estimator with non-negative weights violates the locality property, it is inconsistent.

**Lemma 2** (**Local average estimators without locality are inconsistent**). *In the setting given by Eq. (2.1), let $\widehat{\eta}_n$ be a local average estimator with non-negative weights $W_{n,i}$. Suppose that $\widehat{\eta}_n$ satisfies the UW-property (Def. 2). Assume furthermore that $\widehat{\eta}_n$ does not satisfy locality (Def. 4). Then, there exists a continuous bounded regression function $\eta : [0, 1]^d \to \mathbb{R}$ such that $\widehat{\eta}_n$ is not consistent.*

This result is a straightforward application of Prop. 6 of Stone (1977). Intuitively, when locality is violated, a local average estimator can be highly biased when the regression function $\eta$ has a large amount of local variability. Note that the data models on which we prove locality is necessary in Lemma 2 are more restricted in comparison to that of diversity.

## 4   Inconsistency of random forests

Viewing forests as a special type of local average estimators, we obtain several inconsistency results by considering the choice of subsampling rate in two extreme scenarios: in Section 4.1, we study trees without subsampling, and in Section 4.2, we study trees with constant subsample sizes.

### 4.1   Forests without subsampling can be inconsistent

In this section, we establish inconsistency of some random forests by showing that they violate the diversity condition. In particular, we focus on infinite random forests with the following tree-construction strategy:

**Definition 5** (**Totally randomized deep trees**). *We say a random forest has totally randomized deep trees if its base trees (i) have the UW-property (Def. 2), (ii) are deep (Def. 1), and (iii) are grown on the entire dataset (no subsampling).*

This parameter setup is similar to the one suggested by Geurts et al. (2006), and the term "totally randomized" in Def. 5 follows the naming convention therein.

**Trees with nearest-neighbor-preserving property.** Besides serving as the base algorithms for random forests, spatial partitioning trees are also widely used for other important tasks such as nearest-neighbor search (Yianilos, 1993). We show that, surprisingly, trees that are good for nearest-neighbor search can lead to inconsistent forests when we adopt the parameter setup that is widely used in the random forest community. Given $X_{[n]}$ and any $x \in [0, 1]^d$, we let $X_{(i)}(x)$ denote the $i$-th nearest neighbor of $x$ from the set $\{X_{[n]}\}$ for the Euclidean distance. We define the *nearest-neighbor-preserving* property of a tree as follows.

**Definition 6** (**Nearest-neighbor-preserving property**). *Let $A(\cdot)$ be the routing function associated with a generic (randomized) tree. We say that the tree has nearest-neighbor-preserving property if there exists $\varepsilon > 0$ such that, $\mathbb{P}\left(X_{(1)}(X) \in A(X)\right) \geq \varepsilon$ for infinitely many $n$.*

Intuitively, Def. 6 means that if we route a query point $x$ through the tree to its leaf cell $A(x)$, then its nearest neighbor is likely to be in the same cell, which is quite appealing when trees are used for nearest-neighbor search. However, via Lemma 1, we can now show that such trees lead to inconsistent forests whenever we grow the trees deep and disable subsampling.

**Theorem 1** (**Forests with deep, nearest-neighbor-preserving trees are inconsistent**). *Suppose that the data distribution satisfies the condition in Eq* (2.1). *Suppose that the infinite random forest $\widehat{\eta}_{n,\mathbb{V}_\infty}$ is built with totally randomized deep trees that additionally satisfy the nearest-neighbor-preserving property, Def. 6. Then $\widehat{\eta}_{n,\mathbb{V}_\infty}$ is $\mathbb{L}^2$-inconsistent.*

The intuition behind Theorem 1 is that trees with nearest-neighbor-preserving property are highly homogeneous when subsampling is disabled: given a query point $x$, each tree in the forest tends to retrieve in its leaf of $x$ a very similar set from the training data, namely those data points that are likely nearest neighbors of $x$. This in turn implies violation of diversity and leads to overfitting (and inconsistency) of the random forest.

Theorem 1 suggests that without subsampling, forests of totally randomized trees can still overfit (that is, subsampling is necessary for some forests to be consistent under the totally randomized deep tree construction regime). On the other hand, we speculate that proper subsampling can make the forests consistent again, while fixing other parameters (that is, subsampling is also sufficient for forests consistency here): with subsampling, the nearest-neighbor-preserving property of the base tree algorithm should still hold, but each time applied on a subsample of the original data; taken together, all nearest neighbors on different subsamples are a much more significant set, hence diversity should work again. If this can be proved, then it would imply that, in contrary to common belief (Geurts et al., 2006), different ways of injecting randomness in the tree construction phase may not be equivalent in reducing overfitting, and that subsampling may be more effective than other ways of injecting randomness to the algorithm. We leave this for future work.

**Example: Forests of deep random projection trees.** Random-projection trees (Dasgupta and Freund, 2008) are a popular data structure, both for nearest-neighbor search (Dasgupta and Sinha, 2015) and regression. In particular in the latter case, random-projection tree based estimators were theoretically shown to be $\mathbb{L}^2$-consistent, with a convergence rate that adapts to the intrinsic data dimension for regression problems when they are pruned cleverly (Kpotufe and Dasgupta, 2012). Below we show, however, that two variants of these trees, namely random projection trees (Algorithm 1) and randomized spill trees (Algorithm 2) can make bad candidates as base trees for random forests when tree pruning and data subsampling are disabled.

**Theorem 2** (**Forests of deep random projection trees are inconsistent**). *Suppose that $X$ is distributed according to a measure $\mu$ that has doubling dimension $d_0 \geq 2$. Suppose additionally that the responses satisfy Eq.* (2.1). *Let $c_0$ be a constant such that Dasgupta and Sinha (2015, Theorem 7) holds—we recall this result as Theorem 5 in the Appendix. For any $\delta \in (0, 1/3)$ and $\varepsilon \in (0, 1)$, suppose that we grow the base trees such that each leaf contains at most $n_0$ sample points, where $n_0$ is a constant which does not depend on $n$ and is defined as follows:*

- *(Random projection tree)* $n_0 = \max\left\{ 8 \log 1/\delta \left( \frac{2c_0 d_0^2}{1-\varepsilon} \right)^{d_0}, \exp\left( \frac{2c_0 d_0^3 (8 \log 1/\delta)^{1/d_0}}{1-\varepsilon} \right) \right\}.$

- *(Randomized spill tree)* $n_0 = 8 \log 1/\delta \left( \frac{c_0 d_0}{\alpha(1-\varepsilon)} \right)^{d_0}$, *with* $\alpha \leq \alpha_0 = \alpha_0(c_0, d_0, \varepsilon, \delta).$

*Then the random forest estimator $\widehat{\eta}_{n,\mathbb{V}_\infty}$ is $\mathbb{L}^2$–inconsistent.*

Theorem 2 is a direct consequence of Theorem 1 and Theorem 5; the latter shows that both Algorithms 1 and 2 are nearest-neighbor-preserving.

**Trees with fast shrinking cell diameter.** Local average estimators such as k-nearest-neighbor (k-NN), kernel, and tree based estimators, often make predictions based on information in a neighborhood around the query point. In all these methods, the number of training data contained in

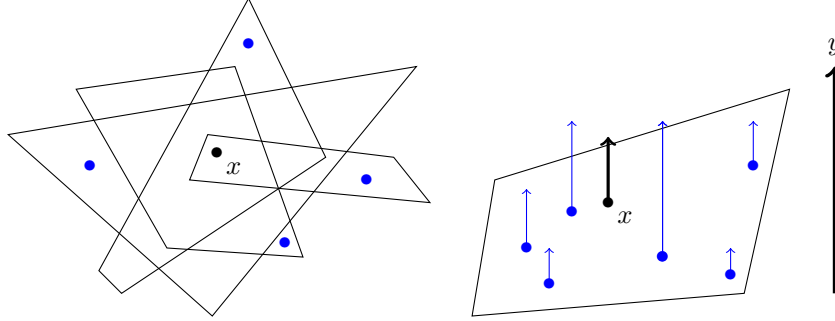

Figure 1: **Left**: Illustration of the "aggregating" effect of a forest induced local neighborhood; the black dot is a query point $x$; the blue points are training points; each cell is the leaf cell of a single tree in the forest containing $x$; the maximal leaf size is $n_0 = 1$. We can see that the aggregated cell (the union of the individual cells) is much larger (less local) than the individual cells. **Right**: The vertical blue lines represent the response values of the sample points belonging to the same cell as the query $x$. The predicted value (in black) is the empirical mean of theses values.

the local neighborhood controls the bias-variance trade-off of the estimator (Devroye et al., 1996, Sec. 6); for these methods to be consistent, the local neighborhood needs to adapt to the training size. For example, in k-NN methods, the size of the neighborhood is determined by the choice of $k$, the number of nearest neighbors of the query point. The classical result of Stone (1977) shows that the k-NN classifier is universally consistent if $k$ grows with $n$ and, at the same time, if $k$ does not grow too fast, namely $k/n \to 0$. We now present a necessary condition on the local neighborhood size for random forests to be consistent. In a particular tree $j$, the local neighborhood of a query $x$ is the leaf cell containing it, $A^j(x)$. In a forest, the local neighborhood of a query can be viewed as an *aggregation* of all possible realizations of tree cells containing $x$.

Intuitively, the aggregated cell in the forest should behave better in the following sense: Consider trees that are fully grown, that is each leaf cell contains only one point. Then the local neighborhood of any query is too small and will result in a tree-based estimator with high variance. Considering the forest, different tree realizations will partition the space differently. This means when fixing a query point $x$, different training data will end up in the leaf cell containing $x$ in different trees, and the aggregated cell can potentially be much larger than the individual tree cell. See the left panel of Fig. 1 for an illustration of this effect. Based on this observation, one would hope that even forests of deep trees can have low enough variance and eventually become consistent.

Our result implies that whether the intuition above holds or not depends on the size of the local neighborhood, controlled by the diameter of the generic (random) function $A(\cdot)$: if the generic tree cell is too small, compared to the data size, then aggregating tree cells will not do much better.

**Proposition 1 (Forests of fully-grown trees with fast shrinking cells are inconsistent).** *Suppose that the data satisfy Eq. (2.1). Suppose additionally that (i) the distribution of $X$ has a density $f$ with respect to the Lebesgue measure on $[0,1]^d$, (ii) there exists constants $f_{\min}$ and $f_{\max}$ such that $\forall x \in [0,1]^d$, $0 < f_{\min} \leq f(x) \leq f_{\max} < +\infty$. Consider the random forest estimator $\widehat{\eta}_{n,\mathbb{V}_\infty}$ built with totally randomized deep trees, and in addition, each tree leaf contains exactly one data point. If with positive probability with respect to $X$, $X_{[n]}$ and $\Theta$, there exists a deterministic sequence $a_n$ of order $\frac{1}{n^{1/d}}$ such that $\mathrm{diam}\,(A(X)) \leq a_n$, then $\widehat{\eta}_{n,\mathbb{V}_\infty}$ is $\mathbb{L}^2$–inconsistent.*

Prop. 1 is similar in spirit to Lin and Jeon (2006, Theorem 3), which is the first result connecting nearest-neighbor methods to random forests. There it was shown that forests with axis-aligned trees can be interpreted to yield sets of "potential nearest neighbors." Using this insight, the authors show that forests of deep axis-aligned trees without subsampling have very slow convergence rate in mean squared error, of order $1/(\log n)^{(d-1)}$, which is much worse than the optimal rate for regression, $O(1/n^{2m/(2m+d)})$ by Stone (1980) (the parameter $m$ controls the smoothness of regression function $\eta$). To the best of our knowledge, this is the only previous result applying to non-artificial data models. We adopt a different approach and directly relate the consistency of forests with the diameter of the generic tree cell. Prop. 1 is stronger than Lin and Jeon (2006), since it establishes

inconsistency, whereas the latter only provides a lower bound on the convergence rate. In addition, Prop. 1 can be applied to any type of trees, including irregularly shaped trees, whereas the aforementioned result is only applicable to axis-aligned trees.

## 4.2 Forests with too severe subsampling can be inconsistent

In contrast to the "totally randomized tree" setup considered in Section 4.1, where subsampling is disabled, we now consider forests with severe subsampling—when the subsample size remains constant as the data size grows to infinity.

**Theorem 3** (**Forests of undersampled fully-grown trees can be inconsistent**). *Suppose that the data satisfy Eq. (2.1) and that $X$ has bounded density. Suppose that the random forest estimator $\widehat{\eta}_{n,\mathbb{V}_\infty}$ has base trees that satisfy the following properties:*

- *Finite subsample size: each tree is constructed on a subsample (sampling with replacement, that is, bootstrapping) of the data $S$ of size $m$, such that $m$ does not vary with $n$;*

- *Fully-grown tree: each tree leaf has exactly one data point.*

*Then $\widehat{\eta}_{n,\mathbb{V}_\infty}$ is $\mathbb{L}^2$–inconsistent.*

Theorem 3 applies Lemma 5 in the undersampled setup. The intuition here is that when the sample points are too "sparse," some cells will have large size when the tree leaves are non-empty (satisfied when trees are fully-grown). Consequently, when a query point falls into a leaf cell, with high probability, it will be far away from the training data in the same cell, violating locality (see the right panel of Fig. 1). It is interesting to compare this result with Prop. 1, which relates the average diameter of a cell in the randomized tree with the tree diversity.

## 5 Discussion

We have shown that random forests with deep trees with either no subsampling or too much subsampling can be inconsistent. One surprising consequence is that trees that work well for nearest-neighbor search problems can be bad candidates for forests without sufficient subsampling, due to a lack of diversity. Another implication is that even totally randomized trees can lead to overfitting forests, which disagrees with the conventional belief that injecting more "randomness" will prevent trees from overfitting (Geurts et al., 2006). In summary, our results indicate that subsampling plays an important role in random forests and may need to be tuned more carefully than other parameters.

There are interesting future directions to explore: (1) While we consider the extreme case of no subsampling or constant subsample size, it would be interesting to explore whether inconsistency holds in cases in-between. Results in this direction would indicate how to choose the subsampling rate in practice. (2) In our analysis, we first let the number of trees $T$ to infinity, and then analyze the consistency of forests as $n$ grows. In the future, it would also be interesting to study the finer interplay between $T$ and $n$ when both of them grow jointly. (3) Bootstrapping, that is subsampling with replacement with subsample size equal to $n$, is a common practice in random forests. It differs subtly from the no subsampling scheme and has been a matter of debate in the theory community (Biau, 2012). We believe that some of our inconsistency results can be extended to the bootstrap case. For example, consider Theorem 2 in the bootstrap case: one would expect that the nearest neighbor property of random projection trees holds on bootstrapped samples as well (according to the central limit theorem for bootstrapped empirical measure (Gine and Zinn, 1990)); when the bootstrap sample size equals $n$, the setup will thus not differ much from the no-subsampling set up, and inconsistency should follow.

## Acknowledgements

The authors thank Debarghya Ghoshdastidar for his careful proofreading of a previous version of this article. This research has been supported by the German Research Foundation via the Research Unit 1735 "Structural Inference in Statistics: Adaptation and Efficiency" and and the Institutional Strategy of the University of Tübingen (DFG ZUK 63).

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
