[Supplementary Material · RF_neurips2018_camera_supplementary.pdf]

# Appendix

We collect in this Appendix all the proofs of the results presented in the paper. Section A contains the main proofs, while Section B contains technical results used in Section A. For completeness' sake, we recall existing theoretical results mentioned in the paper in Section C. Finally, reference to a concentration result as well as a quick summary of the properties of conditional expectation that are used in Section A are collected in Section D and E.

## A Proofs of the main results

**Proof of Lemma 1.** In the proof, we first obtain a bias–variance decomposition of the mean squared error, and then proceed to lower bound the variance term for infinitely many $n$.

Since the diversity condition does not hold, there exists $\delta > 0$ such that, for infinitely many $n$,

$$\mathbb{E}\left[\sum_{i=1}^n W_{n,i}^2(X)\right] \geq \delta. \tag{A.6}$$

Set $n$ as in Eq. (A.6). We define the auxiliary estimator $\bar{\eta}_n$ as

$$\bar{\eta}_n(x) := \sum_{i=1}^n \eta(X_i) W_{n,i}(x) \qquad \text{for any} \quad x \in [0,1]^d. \tag{A.7}$$

According to Lemma 3,

$$\mathbb{E}\left[|\eta(X) - \widehat{\eta}_n(X)|^2\right] = \mathbb{E}\left[|\eta(X) - \bar{\eta}_n(X)|^2\right] + \mathbb{E}\left[|\bar{\eta}_n(X) - \widehat{\eta}_n(X)|^2\right].$$

We now proceed to lower bound the variance term. First, we condition with respect to $X, X_{[n]}$ and $\Theta$ to obtain

$$\mathbb{E}\left[|\bar{\eta}_n(X) - \widehat{\eta}_n(X)|^2 \Big| X, X_{[n]}, \Theta\right] = \mathrm{Var}\left(\bar{\eta}_n(X) - \widehat{\eta}_n(X) \big| X, X_{[n]}, \Theta\right)$$

$$\text{(Eq. (B.12))}$$

$$= \mathrm{Var}\left(\sum_{i=1}^n (Y_i - \eta(X_i)) W_{n,i}(X) \Big| X, X_{[n]}, \Theta\right)$$

$$\text{(definition of } \bar{\eta}_n \text{ (Eq. (A.7)) and } \widehat{\eta}_n \text{ (Eq. (2.4)))}$$

$$= \mathrm{Var}\left(\sum_{i=1}^n \varepsilon_i W_{n,i}(X) \Big| X, X_{[n]}, \Theta\right)$$

$$\text{(Eq. (2.1))}$$

$$= \sum_{i=1}^n W_{n,i}^2(X) \mathrm{Var}\left(\varepsilon_i \big| X, X_{[n]}, \Theta\right)$$

$$\text{(UW-property + independence of the random variables } \varepsilon_i)$$

$$= \sum_{i=1}^n W_{n,i}^2(X) \mathrm{Var}\left(\varepsilon_i\right)$$

$$\text{(each } \varepsilon_i \text{ is independent from } X, X_{[n]} \text{ and } \Theta)$$

$$\mathbb{E}\left[|\bar{\eta}_n(X) - \widehat{\eta}_n(X)|^2 \Big| X, X_{[n]}, \Theta\right] = \sum_{i=1}^n W_{n,i}^2(X)\sigma^2.$$

By the tower property of the conditional expectation (Prop. 2),

$$\mathbb{E}\left[|\bar{\eta}_n(X) - \widehat{\eta}_n(X)|^2\right] = \mathbb{E}\left[\mathbb{E}\left[|\bar{\eta}_n(X) - \widehat{\eta}_n(X)|^2 \Big| X, X_{[n]}, \Theta\right]\right] = \sigma^2 \mathbb{E}\left[\sum_{i=1}^n W_{n,i}^2(X)\right].$$

Finally, recall that $n$ was chosen such that $\mathbb{E}\left[\sum_{i=1}^{n} W_{n,i}^2(X)\right] \geq \delta$. Thus

$$\mathbb{E}\left[|\overline{\eta}_n(X) - \widehat{\eta}_n(X)|^2\right] \geq \delta\sigma^2\,,$$

and we can conclude. $\qquad\qquad\qquad\qquad\qquad\qquad\qquad\qquad\qquad\qquad\qquad\qquad\qquad\quad$ $\square$

**Proof of Lemma 2.** According to the contrapositive of Prop. 6 in Stone (1977), since we assumed that $\widehat{\eta}_n$ has non-negative weights and does not satisfy the locality condition, there exists a bounded continuous function $\eta : [0,1]^d \to \mathbb{R}$ such that the following does not hold:

$$\sum_{i=1}^{n} W_{n,i}(X)\eta(X_i) \longrightarrow 0 \quad \text{in probability}\,.$$

Thus we can choose $\varepsilon > 0$ and $\delta > 0$ such that

$$\mathbb{P}\left(|\eta(X) - \overline{\eta}_n(X)| \geq \varepsilon\right) \geq \delta\,, \qquad\qquad\qquad\qquad\qquad (\text{A.8})$$

for infinitely many $n$—recall that we defined $\overline{\eta}_n(x) = \sum_{i=1}^{n} W_{n,i}(X)\eta(X_i)$. According to Lemma 3, for any $n$,

$$\mathbb{E}\left[|\eta(X) - \widehat{\eta}_n(X)|^2\right] = \mathbb{E}\left[|\eta(X) - \overline{\eta}_n(X)|^2\right] + \mathbb{E}\left[|\overline{\eta}_n(X) - \widehat{\eta}_n(X)|^2\right]\,.$$

In particular,

$$\mathbb{E}\left[|\eta(X) - \widehat{\eta}_n(X)|^2\right] \geq \mathbb{E}\left[|\eta(X) - \overline{\eta}_n(X)|^2\right]\,.$$

Let $n$ be such that Eq. (A.8) holds. Then

$$\begin{aligned}
\mathbb{E}\left[|\eta(X) - \widehat{\eta}_n(X)|^2\right] &\geq \mathbb{E}\left[|\eta(X) - \overline{\eta}_n(X)|^2\right] \\
&\geq \mathbb{P}\left(|\eta(X) - \overline{\eta}_n(X)| \geq \varepsilon\right)\varepsilon^2 \\
&\qquad\qquad\qquad\qquad\qquad \text{(Markov's inequality)} \\
\mathbb{E}\left[|\eta(X) - \widehat{\eta}_n(X)|^2\right] &\geq \delta\varepsilon^2\,. \\
&\qquad\qquad\qquad\qquad\qquad\qquad\quad (\text{Eq. (A.8)})
\end{aligned}$$

Since the last display holds for infinitely many $n$, we can conclude. $\qquad\qquad\qquad\qquad$ $\square$

**Proof of Theorem 1.** Note that the first assumption of Lemma 1 is satisfied. The major part of the proof is to show that the second assumption of Lemma 1, namely Eq. (A.6), is also satisfied.

In this proof, we write $W_{n,i}(X)$ short for $W_{n,i}^{\infty}(X)$. Let $n \in \mathbb{N} \setminus \{0\}$ be as in the nearest-neighbor property. By the definition of the asymptotic weights and the deep tree assumption, for any $1 \leq i \leq n$,

$$W_{n,i}(X) = \mathbb{E}_\Theta\left[\frac{\mathbb{1}_{X_i \in A(X)}}{N(A(X))}\right] \geq \mathbb{E}_\Theta\left[\frac{\mathbb{1}_{X_i \in A(X)}}{n_0}\right]\,. \qquad\qquad (\text{A.9})$$

Let us denote by $W_{n,(1)}(X)$ the asymptotic weight corresponding to the nearest-neighbor of $X$. Since

$$\sum_{i=1}^{n} W_{n,i}^2(X) \geq W_{n,(1)}^2(X) \qquad \text{a.s.}\,,$$

we have

$$\begin{aligned}
\mathbb{E}\left[\sum_{i=1}^{n} W_{n,i}^2(X)\right] &\geq \mathbb{E}\left[W_{n,(1)}^2(X)\right] \\
&\geq \frac{1}{n_0^2}\mathbb{E}\left[\left(\mathbb{E}_\Theta\left[\mathbb{1}_{X_{(1)}(X) \in A(X)}\right]\right)^2\right] \\
&\qquad\qquad\qquad\qquad\qquad\qquad (\text{Eq. (A.9)}) \\
&\geq \frac{1}{n_0^2}\left(\mathbb{E}\left[\mathbb{E}_\Theta\left[\mathbb{1}_{X_{(1)}(X) \in A(X)}\right]\right]\right)^2
\end{aligned}$$

$$(x \mapsto x^2 \text{ is convex + Jensen's inequality})$$

$$= \frac{1}{n_0^2} \mathbb{P}\left(X_{(1)}(X) \in A(X)\right)^2$$

$$\mathbb{E}\left[\sum_{i=1}^{n} W_{n,i}^2(X)\right] \geq \frac{\varepsilon^2}{n_0^2}.$$

$$(\text{nearest-neighbor-preserving property})$$

Since $n_0$ does not depend on $n$, the second assumption of Lemma 1 is satisfied for $\delta = \varepsilon^2/n_0^2$ and we can conclude. $\qquad\square$

**Proof of Theorem 2.** The proof of this result relies on Theorem 1. For both the randomized spill tree and the random projection tree, the UW-property is satisfied. Moreover, by assumption, they are deep trees with almost surely at most $n_0$ sample points per leaf. Thus we only have to check that the nearest-neighbor-preserving property is satisfied, which we achieve thanks to Theorem 5 with $k = 1$.

We first focus on the randomized spill tree case. Let us fix $x \in [0,1]^d$, $\delta \in (0,1/3)$, and $\varepsilon \in (0,1)$. The hypotheses of Theorem 5 are satisfied: provided that $1 \leq \alpha n_0/2$, there is an event $E$ with probability greater than $1 - 3\delta$ such that

$$\mathbb{P}\left(X_{(1)}(x) \notin A(x) \middle| E\right) \leq \frac{c_0 d_0}{\alpha}\left(\frac{8\log 1/\delta}{n_0}\right)^{1/d_0}.$$

By definition of $n_0$, $1 \leq \alpha n_0/2$ holds for any $\alpha$ such that

$$\alpha \leq (4\log 1/\delta)^{d_0-1}\left(\frac{c_0 d_0}{1-\varepsilon}\right)^{\frac{d_0}{d_0-1}} =: \alpha_0\,,$$

and in this case, we have $\mathbb{P}\left(X_{(1)}(x) \notin A(x) \middle| E\right) \leq 1 - \varepsilon$. Since the previous statement is true for any $x \in [0,1]^d$, we have in fact proved that

$$\mathbb{P}\left(X_{(1)}(X) \in A(X) \middle| E\right) \geq \varepsilon\,.$$

Now, since $\mathbb{P}(A|B)\,\mathbb{P}(B) \leq \mathbb{P}(A)$ for any events $A$ and $B$, we obtain

$$\mathbb{P}\left(X_{(1)}(X) \in A(X)\right) \geq \mathbb{P}\left(X_{(1)}(X) \in A(X) \middle| E\right)\mathbb{P}(E) \geq \varepsilon(1 - 3\delta) > 0\,.$$

In other words, the nearest-neighbor-preserving property of Theorem 1 is satisfied and we can conclude.

The proof for random projection trees is similar, with the difference that we have to check whether $1 \leq c_0 3^{d_0}\log 1/\delta$. This is true since $d_0 \geq 2$, $\delta \in (0,1/3)$ and one can take $c_0 \geq 1$ in the statement of Theorem 5. Then, with $E$ defined as before, according to Theorem 5,

$$\mathbb{P}\left(X_{(1)}(x) \notin A(x) \middle| E\right) \leq c_0 d_0(d_0 + \log n_0)\left(\frac{8\log 1/\delta}{n_0}\right)^{1/d_0}.$$

Now, $n_0 \geq 8\log 1/\delta\left(\frac{2c_0 d_0^2}{1-\varepsilon}\right)^{d_0}$, therefore

$$c_0 d_0^2\left(\frac{8\log 1/\delta}{n_0}\right)^{1/d_0} \leq \frac{1-\varepsilon}{2}\,.$$

Moreover, it also holds that $n_0 \geq \exp\left(\frac{2c_0 d_0^3(8\log 1/\delta)^{1/d_0}}{1-\varepsilon}\right)$. Thus

$$c_0 d_0 \log n_0 \left(\frac{8\log 1/\delta}{n_0}\right)^{1/d_0} = c_0 d_0^2 \frac{\log n_0^{1/d_0}}{n_0^{1/d_0}}(8\log 1/\delta)^{1/d_0}$$

$$\leq c_0 d_0^2 \frac{1}{\log n_0^{1/d_0}}(8\log 1/\delta)^{1/d_0}$$

$$(\log x/x \leq 1/\log x \text{ for any } x > 1)$$

$$\leq c_0 d_0^2 \frac{1 - \varepsilon}{2 c_0 d_0^2 \left(8 \log 1/\delta\right)^{1/d_0}} \left(8 \log 1/\delta\right)^{1/d_0}$$

$$c_0 d_0 \log n_0 \left(\frac{8 \log 1/\delta}{n_0}\right)^{1/d_0} \leq \frac{1 - \varepsilon}{2}.$$

We deduce

$$\mathbb{P}\left(X_{(1)}(x) \notin A(x) \big| E\right) \leq \frac{1 - \varepsilon}{2} + \frac{1 - \varepsilon}{2} = 1 - \varepsilon.$$

We conclude the proof with the same argument used in the randomized spill trees case. $\qquad\square$

**Proof of Prop. 1.** In this proof we write $W_{n,i}(X)$ short for $W_{n,i}^{\infty}(X)$. We are going to use Lemma 1 to show that $\widehat{\eta}_{n,\mathbb{V}_\infty}$ is inconsistent.

For any $n \in \mathbb{N} \setminus \{0\}$, it holds that $\sum_{i=1}^{n} W_{n,i}(X) = 1$ almost surely since each cell contains exactly one sample point. Let $(a_n)_{n \geq 1}$ be a deterministic sequence such that $\mathrm{diam}\left(A(X)\right) \leq a_n$ holds with probability greater than $\eta \in (0, 1)$. Set $\delta = \eta/2$ and define $N$ as in Lemma 4. Let $n \geq N$. We have

$$\mathbb{E}\left[\sum_{i=1}^{n} W_{n,i}(X)\, \mathbb{1}_{\mathrm{diam}(A(X)) \leq a_n}\right] = \mathbb{P}\left(\mathrm{diam}\left(A(X)\right) \leq a_n\right) \geq \eta.$$

On the event $\left\{\mathrm{diam}\left(A(X)\right) \leq a_n\right\}$, for any $1 \leq i \leq n$, then $\|X_i - X\| > a_n$ implies $\|X_i - X\| > \mathrm{diam}\left(A(X)\right)$. In turn it holds that $X_i \notin A(X)$, i.e, $W_{n,i}(X) = 0$. Therefore,

$$\mathbb{E}\left[\sum_{i=1}^{n} W_{n,i}(X)\, \mathbb{1}_{\mathrm{diam}(A(X)) \leq a_n}\right] = \mathbb{E}\left[\sum_{i=1}^{n} W_{n,i}(X)\, \mathbb{1}_{\mathrm{diam}(A(X)) \leq a_n}\, \mathbb{1}_{\|X_i - X\| \leq a_n}\right]$$

$$\leq \mathbb{E}\left[\sum_{i=1}^{n} W_{n,i}(X)\, \mathbb{1}_{\|X_i - X\| \leq a_n}\right].$$

Thus we have obtained

$$\mathbb{E}\left[\sum_{i=1}^{n} W_{n,i}(X)\, \mathbb{1}_{\|X_i - X\| \leq a_n}\right] \geq \eta.$$

Define $E$ as the event $\left\{\sum_{i=1}^{n} \mathbb{1}_{\|X_i - X\| \leq a_n} \leq N\right\}$. According to the law of total expectation,

$$\mathbb{E}\left[\sum_{i=1}^{n} W_{n,i}(X)\, \mathbb{1}_{\|X_i - X\| \leq a_n}\right] = \mathbb{E}\left[\sum_{i=1}^{n} W_{n,i}(X)\, \mathbb{1}_{\|X_i - X\| \leq a_n}\bigg| E\right] \mathbb{P}\left(E\right)$$

$$+ \mathbb{E}\left[\sum_{i=1}^{n} W_{n,i}(X)\, \mathbb{1}_{\|X_i - X\| \leq a_n}\bigg| E^{\mathsf{c}}\right] \mathbb{P}\left(E^{\mathsf{c}}\right).$$

Thus

$$\mathbb{E}\left[\sum_{i=1}^{n} W_{n,i}(X)\, \mathbb{1}_{\|X_i - X\| \leq a_n}\bigg| E\right] \mathbb{P}\left(E\right) = \mathbb{E}\left[\sum_{i=1}^{n} W_{n,i}(X)\, \mathbb{1}_{\|X_i - X\| \leq a_n}\right]$$

$$- \mathbb{E}\left[\sum_{i=1}^{n} W_{n,i}(X)\, \mathbb{1}_{\|X_i - X\| \leq a_n}\bigg| E^{\mathsf{c}}\right] \mathbb{P}\left(E^{\mathsf{c}}\right)$$

$$\geq \eta - \mathbb{P}\left(E^{\mathsf{c}}\right).$$
$$\left(\sum_{i=1}^{n} W_{n,i}(X)\, \mathbb{1}_{\|X_i - X\| \leq a_n} \leq 1 \text{ almost surely}\right)$$

According to Lemma 4, we have $\mathbb{P}\left(E^{\mathsf{c}}\right) \leq \delta$ and thus

$$\mathbb{E}\left[\sum_{i=1}^{n} W_{n,i}(X)\, \mathbb{1}_{\|X_i - X\| \leq a_n}\bigg| E\right] \geq \frac{\eta}{2}. \qquad (A.10)$$

Now, according to the Cauchy-Schwarz inequality for discrete sequences, conditionally to $E$,

$$\left(\sum_{i=1}^{n} W_{n,i}(X)\,\mathbb{1}_{\|X_i-X\|\le a_n}\right)^2 \le \sum_{i=1}^{n} W_{n,i}^2(X)\cdot\sum_{i=1}^{n}\mathbb{1}_{\|X_i-X\|\le a_n} \le N\cdot\sum_{i=1}^{n} W_{n,i}(X)^2\,. \quad \text{(A.11)}$$

We write

$$\mathbb{E}\left[\sum_{i=1}^{n} W_{n,i}^2(X)\right] \ge \mathbb{E}\left[\sum_{i=1}^{n} W_{n,i}^2(X)\,\middle|\,E\right]\mathbb{P}(E)$$

(law of total expectation + monotony)

$$\ge \frac{1}{N}\mathbb{E}\left[\left(\sum_{i=1}^{n} W_{n,i}(X)\,\mathbb{1}_{X_i-X\le a_n}\right)^2\middle|\,E\right]\mathbb{P}(E)$$

(Eq. (A.11))

$$\ge \frac{1}{N}\left(\mathbb{E}\left[\sum_{i=1}^{n} W_{n,i}(X)\,\mathbb{1}_{\|X_i-X\|\le a_n}\middle|\,E\right]\right)^2\mathbb{P}(E)$$

($t \mapsto t^2$ convex + conditional Jensen's inequality)

$$\mathbb{E}\left[\sum_{i=1}^{n} W_{n,i}^2(X)\right] \ge \frac{1}{N}\cdot\frac{\eta^2}{4}\cdot(1-\eta/2)\,.$$

(Eq. (B.12))

Since $N$ only depends on quantities that are fixed with respect to $n$, we can conclude thanks to Lemma 1. $\qquad\square$

**Proof of Theorem 3.** The sketch of the proof is the following. First, we use Lemma 5 to find a radius $\rho$ that violates the locality condition for any subsample of the original data. This radius depends on $m$, the size of this subsample. But since $m$ is constant by assumption, $\rho$ violates the locality condition for any $n$. Finally we conclude with Lemma 2.

Let $\varepsilon \in (0,1)$. Set

$$\rho := \frac{1}{2}\left[\frac{(1-\varepsilon)\Gamma\left(\frac{d}{2}+1\right)}{m f_{\max}\pi^{d/2}}\right]^{1/d}\,.$$

Note that $\rho$ does not depend on $n$. To any subset $S \subseteq \{1,\dots,n\}$ corresponds the local average estimator $\widehat{\eta}_m^S$ build upon $(X_i)_{i\in S}$. We denote by $W_{m,i}^S$ its weights. We extend this notation to $W_{n,i}^S = W_{m,i}^S$ if $i \in S$ and $W_{n,i}^S = 0$ otherwise. According to Lemma 5, it holds that

$$\mathbb{E}\left[\sum_{i=1}^{n} W_{n,i}^S(X)\,\mathbb{1}_{\|X_i-X\|\ge\rho}\right] \ge \varepsilon\,.$$

Then, since the weights corresponding to $\widehat{\eta}_n$ satisfy $W_{n,i} = \mathbb{E}\left[W_{n,i}^S\right]$ (where the expectation is with respect to the subsampling), it holds that

$$\mathbb{E}\left[\sum_{i=1}^{n} W_{n,i}(X)\,\mathbb{1}_{\|X_i-X\|\ge\rho}\right] \ge \varepsilon\,.$$

We conclude with Lemma 2. $\qquad\square$

## B  Auxiliary results

In this section, we collect some auxiliary results used in the proofs throughout this paper.

Our first result is a standard bias-variance decomposition used in the proof of Lemma 1 and Lemma 2.

**Lemma 3** (**Bias-variance decomposition**). *Suppose that the observations satisfy Eq. (2.1). Then, for any local average estimator $\widehat{\eta}_n$ satisfying the UW-property,*

$$\mathbb{E}\left[|\eta(X) - \widehat{\eta}_n(X)|^2\right] = \mathbb{E}\left[|\eta(X) - \overline{\eta}_n(X)|^2\right] + \mathbb{E}\left[|\overline{\eta}_n(X) - \widehat{\eta}_n(X)|^2\right].$$

*Proof.* Let $n$ be an integer. We first decompose the mean squared error as

$$
\begin{aligned}
\mathbb{E}\left[|\eta(X) - \widehat{\eta}_n(X)|^2\right] &= \mathbb{E}\left[|\eta(X) - \overline{\eta}_n(X) + \overline{\eta}_n(X) - \widehat{\eta}_n(X)|^2\right] \\
&= \mathbb{E}\left[|\eta(X) - \overline{\eta}_n(X)|^2\right] + \mathbb{E}\left[|\overline{\eta}_n(X) - \widehat{\eta}_n(X)|^2\right] \\
&\quad + 2\,\mathbb{E}\left[(\eta(X) - \overline{\eta}_n(X))\,(\overline{\eta}_n(X) - \widehat{\eta}_n(X))\right].
\end{aligned}
$$

Further inspection of the double-product term shows that

$$
\mathbb{E}\left[(\eta(X) - \overline{\eta}_n(X))\,(\overline{\eta}_n(X) - \widehat{\eta}_n(X))\right] = \mathbb{E}\left[\mathbb{E}\left[(\eta(X) - \overline{\eta}_n(X))\,(\overline{\eta}_n(X) - \widehat{\eta}_n(X))\big|X, X_{[n]}, \Theta\right]\right]
$$
(tower property)

$$
= \mathbb{E}\left[(\eta(X) - \overline{\eta}_n(X))\,\mathbb{E}\left[\overline{\eta}_n(X) - \widehat{\eta}_n(X)\big|X, X_{[n]}, \Theta\right]\right].
$$
($\eta(X)$ and $\overline{\eta}_n(X)$ are $\sigma(X, X_{[n]}, \Theta)$-measurable by the UW-property)

Additionally,

$$
\begin{aligned}
\mathbb{E}\left[\overline{\eta}_n(X) - \widehat{\eta}_n(X)\big|X, X_{[n]}, \Theta\right] &= \overline{\eta}_n(X) - \mathbb{E}\left[\widehat{\eta}_n(X)\big|X, X_{[n]}, \Theta\right] \\
&\quad (\overline{\eta}_n(X) \text{ is } \sigma(X, X_{[n]}, \Theta)\text{-measurable by the UW-property}) \\
&= \sum_{i=1}^{n} \eta(X_i)W_{n,i}(X) - \mathbb{E}\left[\sum_{i=1}^{n} W_{n,i}(X)Y_i\bigg|X, X_{[n]}, \Theta\right] \\
&\quad (\text{definition of } \overline{\eta}_n \text{ (Eq. (A.7)) and } \widehat{\eta}_n \text{ (Eq. (2.4))}) \\
&= \sum_{i=1}^{n} \eta(X_i)W_{n,i}(X) - \sum_{i=1}^{n} \mathbb{E}\left[W_{n,i}(X)Y_i\big|X, X_{[n]}, \Theta\right]
\end{aligned}
$$
(linearity)

$$
\mathbb{E}\left[\overline{\eta}_n(X) - \widehat{\eta}_n(X)\big|X, X_{[n]}, \Theta\right] = \sum_{i=1}^{n} W_{n,i}(X)\left\{\eta(X_i) - \mathbb{E}\left[Y_i\big|X, X_{[n]}, \Theta\right]\right\}.
$$
($W_{n,i}(X)$ is $\sigma(X, X_{[n]}, \Theta)$-measurable by the UW-property)

By irrelevance of independent information (Prop. 2),

$$\mathbb{E}\left[Y_i\big|X, X_{[n]}, \Theta\right] = \mathbb{E}\left[Y_i|X_i\right],$$

and by Eq. (2.1), $\mathbb{E}\left[Y_i|X_i\right] = \eta(X_i)$. We conclude that

$$\mathbb{E}\left[\overline{\eta}_n(X) - \widehat{\eta}_n(X)\big|X, X_{[n]}, \Theta\right] = 0, \tag{B.12}$$

and therefore the double-product term vanishes. We have obtained the following decomposition for the mean squared error:

$$\mathbb{E}\left[|\eta(X) - \widehat{\eta}_n(X)|^2\right] = \mathbb{E}\left[|\eta(X) - \overline{\eta}_n(X)|^2\right] + \mathbb{E}\left[|\overline{\eta}_n(X) - \widehat{\eta}_n(X)|^2\right].$$

$\square$

The following result is used in the proof of Prop. 1 to control the number of sample points falling in a certain ball around $X$.

**Lemma 4** (**Controlling the number of sample points near** $X$). *Let $\delta \in (0, 1/2)$. Under the assumptions of Lemma 1, we can choose constants $0 < m < M < +\infty$ such that, for any $n \in \mathbb{N} \setminus \{0\}$, $m \le a_n n^{1/d} \le M$. Set*

$$C := \frac{\Gamma\left(\frac{d}{2} + 1\right)\log\frac{4}{\delta}}{m f_{\min}}\left(1 + \sqrt{1 + 2m f_{\min}}\right),$$

$$N_0 := \frac{(C+1)M f_{\max} \pi^{d/2}}{\Gamma\left(\frac{d}{2}+1\right)} \quad and \quad N_1 := \left(\frac{8M f_{\max} d}{\delta}\right)^d .$$

*Then, for any $n \geq N := \max(N_0, N_1)$,*

$$\mathbb{P}\left(\sum_{i=1}^{n} \mathbb{1}_{\|X_i - X\| \leq a_n} > N\right) \leq \delta .$$

*Proof.* Set $\partial$ the boundary of $[0,1]^d$. We first show that for any fixed $x \in [0,1]^d$ far away from the boundary, that is, $x$ such that $d(x, \partial) \geq a_n$, then

$$\mathbb{P}\left(\sum_{i=1}^{n} \mathbb{1}_{\|X_i - x\| \leq a_n} > N\right) \leq \delta/2 .$$

Set $x \in [0,1]^d$ such that $d(x, \partial) \geq a_n$ and $p := \mu(\mathcal{B}(x, a_n))$. We write

$$\mathbb{P}\left(\sum_{i=1}^{n} \mathbb{1}_{\|X_i - x\| \leq a_n} > N\right) \leq \mathbb{P}\left(\sum_{i=1}^{n} \mathbb{1}_{\|X_i - x\| \leq a_n} > N_0\right)$$

$$= \mathbb{P}\left(\frac{1}{n}\sum_{i=1}^{n} \mathbb{1}_{\|X_i - x\| \leq a_n} - p > \frac{N_0}{n} - p\right)$$

$$\leq \mathbb{P}\left(\left|\frac{1}{n}\sum_{i=1}^{n} \mathbb{1}_{\|X_i - x\| \leq a_n} - p\right| > \frac{N_0}{n} - p\right)$$

We notice that

$$p = \mu(\mathcal{B}(x, a_n))$$

(definition of $\mu$)

$$\geq f_{\min} \mu_{\text{Leb}}\left(\mathcal{B}(x, a_n) \cap [0,1]^d\right)$$

($\mu$ has bounded density on $[0,1]^d$)

$$= f_{\min} \mu_{\text{Leb}}(\mathcal{B}(x, a_n))$$

(we assumed $d(x, \partial) \geq a_n$)

$$= \frac{f_{\min} \pi^{d/2} a_n^d}{\Gamma\left(\frac{d}{2}+1\right)}$$

(volume of the hypersphere in dimension $d$)

$$p \geq \frac{m f_{\min} \pi^{d/2}}{n \Gamma\left(\frac{d}{2}+1\right)},$$

($a_n \geq m/n^{1/d}$)

where $\mu_{\text{Leb}}$ is the Lebesgue measure on $\mathbb{R}^d$. The converse direction is similar, and we write

$$\frac{m f_{\min} \pi^{d/2}}{n \Gamma\left(\frac{d}{2}+1\right)} \leq p \leq \frac{M f_{\max} \pi^{d/2}}{n \Gamma\left(\frac{d}{2}+1\right)} . \tag{B.13}$$

Therefore,

$$\frac{N_0}{(C+1)n} = \frac{M f_{\max} \pi^{d/2}}{\Gamma\left(\frac{d}{2}+1\right) n} \geq p ,$$

and we deduce that $N_0/n - p > pC$. As a consequence,

$$\mathbb{P}\left(\sum_{i=1}^{n} \mathbb{1}_{\|X_i - x\| \leq a_n} > N\right) \leq \mathbb{P}\left(\left|\frac{1}{n}\sum_{i=1}^{n} \mathbb{1}_{\|X_i - x\| \leq a_n} - p\right| > pC\right) .$$

Set

$$Z_n := \frac{1}{n}\sum_{i=1}^{n}\mathbb{1}_{\|X_i - x\| \le a_n} \ .$$

The random variable $Z_n$ is a normalized sum of independent 0–1-valued Bernoulli random variables taking value 1 with probability $p$. Note that $\sum_i \mathbb{E}\left[\mathbb{1}_{\|X_i-x\|\le a_n}^2\right] = np$. According to the Bernstein's inequality (Lemma 6), our choice of $C$ and the lower bound on $p$,

$$\mathbb{P}\left(|Z_n - p| > pC\right) \le 2\exp\left(-\frac{nC^2p}{2 + 2C/3}\right) \le \frac{\delta}{2} \ .$$

We have proved that, for any fixed $x$ such that $d\left(x, \partial\right) \ge a_n$,

$$\mathbb{P}\left(\sum_{i=1}^{n}\mathbb{1}_{\|X_i-x\|\le a_n} > N\right) \le \delta/2 \ .$$

We now focus on the points that are near the boundary of $[0,1]^d$. Since we assumed that $X$ has bounded density on $[0,1]^d$, it holds that

$$\mathbb{P}\left(d\left(X,\partial\right) \le a_n\right) \le f_{\max}\mu_{\text{Leb}}\left(\left\{x \in [0,1]^d \text{ s.t. } d\left(x, \partial\left([0,1]^d\right)\right) \le a_n\right\}\right)$$

$$\le f_{\max} \cdot 4d \cdot a_n$$

$$\text{(the unit cube has } 2d\,(d-1)\text{-dimensional faces)}$$

$$\le \frac{4Mf_{\max}d}{n^{1/d}}$$

$$(a_n \le M/n^{1/d})$$

$$\mathbb{P}\left(d\left(X,\partial\right) \le a_n\right) \le \frac{\delta}{2} \ ,$$

$$(n \ge N_1)$$

and we can conclude. $\qquad\square$

**Lemma 5** (**Relation between locality and sample size**). *Suppose that the data satisfy Eq. (2.1) and that $X$ has bounded density. Consider an infinite random forest estimator $\widehat{\eta}_n$ whose base trees satisfy the two properties listed in Theorem 3. Let $\varepsilon \in (0,1)$. Then, for any*

$$\rho < \left[\frac{(1-\varepsilon)\Gamma\left(\frac{d}{2}+1\right)}{nf_{\max}\pi^{d/2}}\right]^{1/d} \ ,$$

*we have*

$$\mathbb{E}\left[\sum_{i=1}^{n}W_{ni}(X)\,\mathbb{1}_{\|X_i-X\|\ge\rho}\right] \ge \varepsilon \ .$$

*Proof.* The intuition behind the proof is very simple: if $\rho$ is small enough with respect to the size of the cells, since $X$ has bounded density on $[0,1]^d$, then it is very unlikely that $X$ falls into balls of radius $\rho$ centered in the sample points—see the Right panel of Fig. 1.

First, we notice that

$$\mathbb{E}\left[\sum_{i=1}^{n}W_{ni}(X)\,\mathbb{1}_{\|X_i-X\|\ge\rho}\right] = \mathbb{E}\left[\sum_{i=1}^{n}\mathbb{E}_{\Theta}\left[\frac{\mathbb{1}_{X_i\in A(X)}}{N(A(X))}\right]\mathbb{1}_{\|X_i-X\|\ge\rho}\right] \ .$$

Since the leaves of the tree contain exactly one data point, the leaves are non-empty, Therefore,

$$\mathbb{E}\left[\sum_{i=1}^{n}W_{ni}(X)\,\mathbb{1}_{\|X_i-X\|\ge\rho}\right] = \mathbb{E}\left[\sum_{i=1}^{n}\mathbb{1}_{X_i\in A(X)}\,\mathbb{1}_{\|X_i-X\|\ge\rho}\right] \ .$$

Again, since $N(A(X)) = 1$ almost surely, we can set unambiguously $A_i$ the cell containing data point $X_i$, and $X_i \in A(X)$ is equivalent to $X \in A_i$. We write

$$\mathbb{P}\left(X_i \in A(X) \text{ and } \|X - X_i\| \geq \rho \big| X_{[n]}, \Theta\right) = \mathbb{P}\left(X \in A_i \text{ and } \|X - X_i\| \geq \rho \big| X_{[n]}, \Theta\right)$$
$$= \mathbb{P}\left(X \in A_i \setminus \mathcal{B}(X_i, \rho) \big| X_{[n]}, \Theta\right).$$

By the union bound,

$$\sum_{i=1}^{n} \mathbb{P}\left(X_i \in A(X) \text{ and } \|X - X_i\| \geq \rho \big| X_{[n]}, \Theta\right) \geq \mathbb{P}\left(X \in \bigcup_{i=1}^{n} A_i \setminus \mathcal{B}(X_i, \rho) \bigg| X_{[n]}, \Theta\right)$$

$$\geq \mathbb{P}\left(X \in [0,1]^d \setminus \bigcup_{i=1}^{n} \mathcal{B}(X_i, \rho) \bigg| X_{[n]}, \Theta\right)$$

$$(\bigcup_i A_i = [0,1]^d)$$

$$\geq 1 - \mathbb{P}\left(X \in \bigcup_{i=1}^{n} \mathcal{B}(X_i, \rho) \bigg| X_{[n]}, \Theta\right)$$

(union bound)

$$\geq 1 - n \cdot \mathbb{P}\left(X \in \mathcal{B}(X_i, \rho) \big| X_{[n]}, \Theta\right)$$

(satisfies the bounded density assumption)

$$\geq 1 - n \cdot f_{\max} \cdot \text{Vol}\left(\mathcal{B}(X_1, \rho)\right)$$

$$\sum_{i=1}^{n} \mathbb{P}\left(X_i \in A(X) \text{ and } \|X - X_i\| \geq \rho \big| X_{[n]}, \Theta\right) \geq 1 - \frac{n f_{\max} \pi^{d/2} \rho^d}{\Gamma\left(\frac{d}{2} + 1\right)}.$$

We deduce that

$$\sum_{i=1}^{n} \mathbb{P}\left(X_i \in A(X) \text{ and } \|X - X_i\| \geq \rho\right) \geq 1 - \frac{n f_{\max} \pi^{d/2} \rho^d}{\Gamma\left(\frac{d}{2} + 1\right)},$$

and we can conclude. $\qquad\square$

$$A(X) = A_i$$

Figure 2: Proof of Lemma 5. The black dots correspond to sample points, the circles around them are of radius $\rho$. The cells form a partition of $[0,1]^2$. Here $X$ belong to the same cell as $X_i$ (in blue).

## C Previous results

**Theorem 4** (**Consistence of local average estimators Stone, 1977, Theorem 1**). *Consider the local average estimator $\widehat{\eta}_n$ defined in Eq. (2.4) and suppose that the following conditions are satisfied.*

1. *There is a $C \geq 1$ such that, for every nonnegative Borel function $f$ on $\mathbb{R}^d$, and for any $n \geq 1$,*

$$\mathbb{E}\left[\sum_{i=1}^n |W_{n,i}(X)| \, f(X_i)\right] \leq C \, \mathbb{E}\left[f(X)\right] .$$

2. *There exists $D \geq 1$ such that $\mathbb{P}\left(\sum_i |W_{n,i}(X)| \leq D\right) = 1$, for all $n \geq 1$;*

3. *$\sum_i |W_{n,i}(X)| \, \mathbb{1}_{\|X_i - X\| > a} \to 0$ in probability for all $a > 0$;*

4. *$\sum_i W_{n,i}(X) \to 1$ in probability;*

5. *$\max_i |W_{n,i}(X)| \to 0$ in probability.*

*Then the local average estimator $\widehat{\eta}_n$ is consistent.*

**Theorem 5** (**Nearest-neighbor search guarantees Dasgupta and Sinha, 2015, Theorem 7**). *There is an absolute constant $c_0$ for which the following holds. Suppose $\mu$ is a doubling measure on $\mathbb{R}^d$ of intrinsic dimension $d_0 \geq 2$, i.e.,*

$$\forall x \in [0,1], \quad \forall r > 0, \quad \forall a \geq 1, \qquad 0 < \mu(\mathcal{B}(x, ar)) < a^{d_0}\mu(\mathcal{B}(x, r)) .$$

*Pick any query $x \in [0,1]^d$ and draw $X_1, \ldots, X_n$ independently from $\mu$. Let $n_0$ be as before the maximal number of sample points in a leaf. For any $\delta \in (0, 1/3)$, with probability at least $1 - 3\delta$ over the choice of data:*

- *For the randomized spill tree, if $k \leq \alpha n_0 / 2$,*

$$\mathbb{P}\left(\text{tree fails to return the } k\text{-nearest neighbors of } x\right) \leq \frac{c_0 d_0 k}{\alpha}\left(\frac{8\max(k, \log 1/\delta)}{n_0}\right)^{1/d_0} .$$

- *For the random projection tree, if $k \leq c_0(3k)^{d_0}\max(k, \log 1/\delta)$,*

$$\mathbb{P}\left(\text{tree fails to return the } k\text{-nearest neighbors of } x\right) \leq c_0 d_0 k(d_0 + \log n_0)\left(\frac{8\max(k, \log 1/\delta)}{n_0}\right)^{1/d_0} .$$

**Theorem 6** (**Convergence w.r.t. number of trees (Scornet, 2016, Theorem 3.1)**). *Define $K_n(\cdot, \cdot) : [0,1]^d \times [0,1]^d \to [0,1]$ the random forest connection function as*

$$K_n(x, y) = \mathbb{P}\left(x \text{ and } y \text{ in the same cell} | \mathcal{D}_n\right) .$$

*Consider a continuous or discrete random forest, that is, assume $K_n$ piecewise-constant or continuous for any fixed $\mathcal{D}_n$. Then, conditionally on the data $\mathcal{D}_n$, for almost every query points $x \in [0,1]^d$, we have*

$$\widehat{\eta}_{n, \mathbb{V}_T}(x) \xrightarrow[T \to +\infty]{} \widehat{\eta}_{n, \mathbb{V}_\infty}(x) .$$

**Theorem 7** (**Infinite forests have smaller risks (Scornet, 2016, Theorem 3.3)**). *Suppose that*

$$Y = \eta(X) + \varepsilon ,$$

*where $\varepsilon$ is a centered Gaussian random variable with finite variance $\sigma^2$, independent of $X$. Assume also that $\|\eta\|_\infty < \infty$. Then, for all $T, n \in \mathbb{N} \setminus \{0\}$,*

$$\mathbb{E}\left[|\widehat{\eta}_{n, \mathbb{V}_T}(X) - \eta(X)|^2\right] = \mathbb{E}\left[|\widehat{\eta}_{n, \mathbb{V}_\infty}(X) - \eta(X)|^2\right] + \frac{1}{T}\mathbb{E}_{X, \mathcal{D}_n}\left[\mathrm{Var}_\Theta\left(\widehat{\eta}_{n, A}(X)\right)\right] .$$

# D    A concentration inequality

The following result is known as the Bernstein's inequality.

**Lemma 6** (**Bernstein's inequality (Boucheron et al., 2013, Eq. (2.10))**). *Let $Z_1, \ldots, Z_n$ be independent random variables. Assume that there exist positive numbers $b$ and $v$ such that*

$$\forall 1 \le i \le n, \ |Z_i| \le b \quad a.s., \quad and \quad \sum_i \mathbb{E}\left[Z_i^2\right] \le v.$$

*Then, for any $t > 0$,*

$$\mathbb{P}\left(\sum_i Z_i - \mathbb{E}\left[Z_i\right] > t\right) \le \exp\left(-\frac{t^2}{2(v + bt/3)}\right).$$

# E    Conditional expectation

In this section, we recall the basic properties of the conditional expectation that are used throughout this paper. We refer to Billingsley (2008, Chapter 6, Section 34) for a proof of the following facts.

**Proposition 2** (**Basic properties of conditional expectation**). *Let $X$ and $Y$ be integrable random variables, let $\mathcal{G}$ and $\mathcal{H}$ be subalgebras of $\mathcal{F}$. Then the following hold:*

1. *(linearity) For any real numbers $\alpha, \beta$,*

$$\mathbb{E}\left[\alpha X + \beta Y | \mathcal{G}\right] = \alpha \mathbb{E}\left[X | \mathcal{G}\right] + \beta \mathbb{E}\left[Y | \mathcal{G}\right] \quad a.s.$$

2. *(monotonicity) If $X \le Y$ a.s., then $\mathbb{E}\left[X | \mathcal{G}\right] \le \mathbb{E}\left[Y | \mathcal{G}\right]$ a.s.*

3. *(conditional Jensen) If $f$ is a convex function such that $f(X)$ is integrable, then*

$$\mathbb{E}\left[f(X) | \mathcal{G}\right] \le f\left(\mathbb{E}\left[X | \mathcal{G}\right]\right) \quad a.s.$$

4. *(measurability) If $Y$ is $\mathcal{G}$-measurable and $XY$ is integrable, then*

$$\mathbb{E}\left[XY | \mathcal{G}\right] = Y \mathbb{E}\left[X | \mathcal{G}\right] \quad a.s.$$

5. *(tower property) If $\mathcal{H} \subseteq \mathcal{G}$, then*

$$\mathbb{E}\left[\mathbb{E}\left[X | \mathcal{G}\right] | H\right] = \mathbb{E}\left[X | \mathcal{H}\right] \quad a.s.$$

6. *(irrelevance of independent information) If $\mathcal{H}$ is independent of $\sigma\left(\mathcal{G}, X\right)$, then*

$$\mathbb{E}\left[X | \sigma\left(\mathcal{G}, \mathcal{H}\right)\right] = \mathbb{E}\left[X | \mathcal{G}\right] \quad a.s.$$

*In particular, if $X$ is independent of $\mathcal{H}$, then $\mathbb{E}\left[X | \mathcal{H}\right] = \mathbb{E}\left[X\right]$ a.s.*