[Reviews · NeurIPS 2018]

Reviewer 1



UPDATE AFTER AUTHORS' RESPONSE I need to clarify: in my opinion, the thing that must be fixed is clarifying the analysis applies to unsupervised RF. My comment about a little lower relevance is to explain why I did not give the paper a higher score. SUMMARY The paper is a theoretical analysis of how data point subsampling impacts unsupervised random forests (unsupervised = tree structure not based on response variable Y). Suppose the forest is grown with an infinite number of trees, using infinite data. It shows that two scenarios produce forests that are asymptotically inconsistent: a) building forests of deep trees without subsampling, and b) building forests with fixed subsample size and fully grown trees (one data point per leaf). REVIEW The question of when random forests fail or do poorly is interesting because those situations seem to be rare in practice. Anecdotally, I can recall only a couple examples from a decade of applied work. Understanding such corner cases improves the community's ability to use these tools scientifically. This paper has three strengths and two weaknesses. Strengths: * This is the first inconsistency analysis for random forests. (Verified by quick Google scholar search.) * Clearly written to make results (mostly) approachable. This is a major accomplishment for such a technical topic. * The analysis is relevant to published random forest variations; these include papers published at ICDM, AAAI, SIGKDD. Weaknesses: * Relevance to researchers and practitioners is a little on the low side because most people are using supervised random forest algorithms. * The title, abstract, introduction, and discussion do not explain that the results are for unsupervised random forests. This is a fairly serious omission, and casual readers would remember the wrong conclusions. This must be fixed for publication, but I think it would be straightforward to fix. Officially, NIPS reviewers are not required to look at the supplementary material. Because of having only three weeks to review six manuscripts, I was not able to make the time during my reviewing. So I worry that publishing this work would mean publishing results without sufficient peer review. DETAILED COMMENTS * p. 1: I'm not sure it is accurate to say that deep, unsupervised trees grown with no subsampling is a common setup for learning random forests. It appears in Geurts et al. (2006) as a special case, sometimes in mass estimation [1, 2], and sometimes in Wei Fan's random decision tree papers [3-6]. I don't think these are used very much. * You may want to draw a connection between Theorem 3 and isolation forests [7] though. I've heard some buzz around this algorithm, and it uses unsupervised, deep trees with extreme subsampling. * l. 16: "random" => "randomized" * l. 41: Would be clearer with forward pointer to definition of deep. * l. 74: "ambient" seems like wrong word choice * l. 81: Is there a typo here? Exclamation point after \thereexists is confusing. * l. 152; l. 235: I think this mischaracterizes Geurts et al. (2006), and the difference is important for the impact stated in Section 4. Geurts et al. include a completely unsupervised tree learning as a special case, when K = 1. Otherwise, K > 1 potential splits are generated randomly and unsupervised (from K features), and the best one is selected *based on the response variable*. The supervised selection is important for low error on most data sets. See Figures 2 and 3; when K = 1, the error is usually high. * l. 162: Are random projection trees really the same as oblique trees? * Section 2.2: very useful overview! * l. 192: Typo? W^2? * l. 197: No "Eq. (2)" in paper? * l. 240: "parameter setup that is widely used..." This was unclear. Can you add references? For example, Lin and Jeon (2006) study forests with adaptive splitting, which would be supervised, not unsupervised. * Based on the abstract, you might be interested in [8]. REFERENCES [1] Ting et al. (2013). Mass estimation. Machine Learning, 90(1):127-160. [2] Ting et al. (2011). Density estimation based on mass. In ICDM. [3] Fan et al. (2003). Is random model better? On its accuracy and efficiency. In ICDM. [4] Fan (2004). On the optimality of probability estimation by random decision trees. In AAAI. [5] Fan et al. (2005). Effective estimation of posterior probabilities: Explaining the accuracy of randomized decision tree approaches. In ICDM. [6] Fan el al. (2006). A general framework for accurate and fast regression by data summarization in random decision trees. In KDD. [7] Liu, Ting, and Zhou (2012). Isolation-based anomaly detection. ACM Transactions on Knowledge Discovery from Data, 6(1). [8] Wager. Asymptotic theory for random forests. https://arxiv.org/abs/1405.0352

Reviewer 2



This paper studies necessary conditions for (infinite) ensembles of randomized regression trees to be inconsistent (in regression). The authors first establish that locality and diversity are necessary conditions for the consistency of forests. From this result, they show that several types of forests are inconsistent: forests of extremely randomized deep trees satisfying the nearest-neighbor preserving property (which includes two random projection tree algorithms as instances), forests of fully-grown trees with fast shrinking cells, and finally forests grown with too severe subsampling. The paper contains interesting and, to the best of my knowledge, original theoretical results about (in)consistency of forests of randomized trees. The originality of the paper is to try to characterize conditions under which generic random forests models are inconsistent, while other papers focus on consistency proof of some simplified algorithms. The paper is also very well written and pleasant to read. The authors very well summarize their main theoretical results in the paper, while detailed proofs are given in the supplementary material. Although the paper is well written, I find however that several statements or implications that are drawn from the theorems in the paper are too strong or not strictly correct. The authors overstate a bit the surprising nature of their results and how much they contradict statements from the literature. Just to give two examples of statements that I found not well argued: 1) About the importance of subsampling: "we show that subsampling of data points during the tree construction phase is critical: forests can become inconsistent with either no subsampling or too severe subsampling", "without subsampling, forests of deep trees can become inconsistent due to violation of diversity", "In summary, our results indicate that subsampling plays an important role in random forests and may need to be tuned more carefully than other parameters". I have two problems with these statements: First, I don't see where it is formally proven (in this paper or in another) that adding subsampling to the different tree models for which inconsistency is proven in the paper will lead to consistency. Actually, page 6, the authors say "we speculate that sufficient subsampling can make the forests consistent again...We leave this for future work". In the absence of a proof, I think the previous statements are premature. Second, even if subsampling would lead to consistency, I don't think subsampling is really crucial per se. Forests of deep trees are inconsistent because the trees are deep. Subsampling is thus clearly not the only or even the most obvious solution. Another solution to make them consistent could be for example to prune the trees (in a way that depends on learning sample size). As subsampling is not the only solution, it is not critical per se, what is critical is the effect that it has on diversity, not its added randomness. 2) About randomness and overfitting: "Theorem 2 implies that forests of extremely randomized trees can still overfit. This results disagrees with the conventional belief that injecting more "randomness" prevents tree from overfitting" and in the conclusion: "Another implication is that even extremely randomized tree can lead to overfitting forests, which disagrees with the conventional belief that injecting more "randomness" will prevent trees from overfitting". I don't understand this statement and the link with the theorem provided in the paper. The conventional belief is not that injecting more randomness implies NO overfitting but rather than injecting more randomness implies LESS overfitting. And I see no violation of this common belief in the paper: forests of extremely randomized deep trees have a low variance but not a zero variance (that's why they are not consistent), and adding subsampling is a way to add even more randomness that can lead to less overfitting and thus consistency, as suggested in the paper. So, I don't see any contradiction or surprising results here. Overall, I think that the authors should be more cautious when discussing the implications of their theorems. This requires in my opinion a significant rewrite of some parts of the paper. The paper topic does not fit also very well to the format of a conference paper. It does not focus on a single clear idea, as the contribution is a series of theorems that remains incremental with respect to previous results in this domain and also does not end any story. Their practical impact is also limited as they concern simplified models (UW trees) that are not really competitive in practice. The paper has also a 11-pages supplementary material that is difficult to check within the time allocated for reviews at conferences. A statistical journal might be a better target for this work. Minor comments: * All theorems refer to a data model in Eq (2) and the authors mention that restrictions on the data model are more important for Lemma 2 than for Lemma 1. Actually, I could not find any Eq (2), neither in the paper nor in the appendix. What is this data model? * The use of the term extremely randomized trees in the paper does not seem to match the use of this name in (Geurts et al., 2006) and is therefore a bit misleading. Extremely randomized trees in (Geurts et al., 2006) are not UW trees because the best split is still selected among K inputs selected at random, which is thus selected on the basis of the labels. Extremely randomized trees are UW trees only when K=1 and in this case, they are called Totally randomized trees in (Geurts et al., 2006). Furthermore, extremely randomized trees in (Geurts et al., 2006) are grown in a specific manner, while Definition 5 is generic. Why not calling them simply "deep UW trees"? Update after the author's reponses I thank the authors for their response. They do not totally address my concerns however: - Importance of subsampling: "The reviewer is also correct in that we did not formally prove but only speculate that with proper subsampling, our RF models shown inconsistent in our paper could become consistent again". Then, I reiterate my comment that you can not yet claim in the paper that subsampling is critical. - Randomness and overfitting: "We agree with the reviewer that the conventional belief is that “injecting more randomness implies less overfitting".This is exactly what we would like to argue against". "We argue (here and in paper) that our Theorem 2 suggests that this type of thinking is incorrect, since highly randomized trees without subsampling can still overfit." Again, to me adding subsampling means injecting more randomness, which implies less overfitting. I still don’t see how the result in theorem 2 really contradicts this statement. I agree however that different ways of injecting randomness are certainly not equivalent and that the RF variant of [Geurts et al., 2006] might overfit more than Breiman’s original Random Forests. But this is not strictly shown in the paper and I don’t really see how this could be proven as it is difficult to compare the levels of randomness of different randomization schemes. Note that the link between overfitting and consistency is not totally clear to me. At finite sample size, a model might overfit less than another, even if only the latter is consistent. Overall, I still believe that the discussion about randomness and overfitting in the paper is confusing and needs to be clarified. Note that I acknowledge the value of the theoretical results in the paper. I still believe however that the authors should be more cautious in the conclusions they draw from these results.

Reviewer 3



UPDATE AFTER AUTHOR RESPONSE: I have read the author response, and I maintain my score. This paper establishes several new theoretical properties of tree ensemble methods. The authors first establish two conditions, diversity and locality, that are necessary for local average estimators in general to be consistent. Viewing tree ensembles as a type of local average estimators, they then show that under certain settings, tree ensembles can be inconsistent by violating the diversity or the locality condition. While Random forests are widely used, their good predictive power still remains unexplained from a theoretical point of view, and the present work is certainly an interesting contribution to the theoretical study of the consistency of Random forests. While most of the (few) previous works focus on deriving settings for which it can be shown that a tree forest is consistent, the present paper rather focuses on negative results by presenting settings under which the forest becomes inconsistent. Theorem 1 shows that without subsampling, randomized, deep trees having the nearest-neighbor-preserving property are always inconsistent. As a consequence, Theorem 2 shows that without subsampling, random projection trees can be inconsistent. Proposition 1 shows that without subsampling, randomized, fully-grown trees with fast shrinking cells diameter can be inconsistent. Finally, Theorem 3 shows that when using too severe subsampling, fully-grown trees are inconsistent. All the claims are supported by theoretical proofs. I spotted a few errors within the proofs that seem to be typos rather than mathematical errors (see below). The paper is well-organized and very clearly written. Here is my main comment: the authors say that the Extremely randomized trees (or Extra-trees) proposed by Geurts et al. (2006) are constructed “without using information from responses Y_[n]” (lines 152-153). This is not true. While at each tree node, K splits are indeed selected completely in a random way, the split that is eventually selected is the one that, among the K random splits, maximises the output variance reduction. The term “Extremely randomized deep trees” used for Definition 5 is therefore quite misleading as the reader might believe that this the definition of the Extra-trees (as actually emphasized by the authors below the definition) while the Extra-trees actually do not have the UW-property. The authors should clarify that and remove any ambiguity. Typos: - Throughout the paper and the appendix, the authors refer to Equation (2) for the data properties, but Equation (2) does not exist. - Equation between lines 452 and 453: the sum should be towards \eta(X) instead of 0. - Equation between lines 477 and 478: the first exponent should be 1/(d0-1) instead of (d0-1). - Line 604, conditional Jensen property: this should be a “greater or equal to” sign.

Reviewer 4



In this paper, the authors proposed sufficient conditions for random forests to be inconsistent. By linking random forests to local average estimators, they showed that a random forests model is consistent only if the diversity and locality conditions hold. The diversity condition requires that training examples have asymptotic zero weights. The locality condition requires that the total training weight outside a fixed ball to be asymptotically 0. Both of the conditions make intuitive sense -- to make an estimator consistent, we need to average over asymptotically infinite amount of data. The authors then continue on to study the properties of specific tree structures. There are a couple points that I think the authors could improve on this paper: 1. The authors should explicitly mention that the tree partition needs to be unsupervised. Although this is implied from the UW-property, where the weights depend only on the unlabeled data, I think the authors should explicitly emphasize that. 2. Although the paper is mostly theoretical, a numerical study is still helpful. The authors could empirically show that as the sample size grows, predictions from random forests that do not satisfy the diversity and localness conditions do not converge to the true value. If the authors need more space, they could shorten Section 2, in which most of the backgrounds should be assumed known by readers. The strength of this paper is it is easy to follow and is interesting from a theoretical and practical point of view. The weakness of the paper is that it lacks a numerical section that empirically examines the authors findings.